# Instrument Context Relevance Evaluation, Translation, and Psychometric Testing of the Diabetes Eating Problem Survey-Revised (DEPS-R) among People with Type 1 Diabetes in China

**DOI:** 10.3390/ijerph18073450

**Published:** 2021-03-26

**Authors:** Wencong Lv, Qinyi Zhong, Jia Guo, Jiaxin Luo, Jane Dixon, Robin Whittemore

**Affiliations:** 1Clinical Nursing Department, Xiangya School of Nursing, Central South University, Changsha 410013, China; lwc_9813@csu.edu.cn (W.L.); zhongqinyi71@csu.edu.cn (Q.Z.); 187812048@csu.edu.cn (J.L.); 2School of Nursing, Yale University, New Haven, CT 06520, USA; jane.dixon@yale.edu (J.D.); robin.whittemore@yale.edu (R.W.)

**Keywords:** diabetes eating problem survey-revised, context relevance, instrument translation, psychometric property, type 1 diabetes

## Abstract

Background: People with type 1 diabetes are susceptible to disordered eating behaviors. The American Diabetes Association recommends using the Diabetes Eating Problem Survey-Revised (DEPS-R) to screen them. There is no validated diabetes-specific screening measure in China. The objectives were to adapt DEPS-R into Mandarin Chinese and to test its psychometric properties among youths and adults with type 1 diabetes in China, respectively. Methods: This study was conducted in two phases. Phase 1 included context relevance evaluation and instrument translation. Phase 2 was psychometric testing of reliability and construct validity among 89 youths (8~17 years old) and 61 adults with type 1 diabetes. Result: The Context Relevance Index and Translation Validity Index of this instrument were good. Strong internal consistency reliability correlations and convergent validity were demonstrated among youths and adults. Discussion: The Chinese version of the DEPS-R is a valid and reliable tool for screening disordered eating behaviors in Chinese youths and adults with type 1 diabetes. The Context Relevance Index is advocated to evaluate the difference between the context in which an instrument was originally developed and the target context.

## 1. Introduction

Type 1 diabetes (T1D) is a chronic autoimmune disease characterized by insulin deficiency [1]. Over the past several decades, the average annual increase of the incidence of T1D has been 3–4% worldwide [2]. According to the international clinical guidelines, keeping a regular and healthy eating habit is a prerequisite for managing diabetes [3,4]. However, it was reported that people with T1D are inherently more prone to issues surrounding food in a systematic review [5], which may be due to abnormal recognition of metabolic signals of people with T1D resulting in subsequent dysregulated eating [6]. A two- to threefold increased prevalence of disordered eating behaviors in individuals with T1D was seen as compared to age and sex-matched people without T1D [5,7]. Disordered eating behaviors generally refer to maladaptive diet-related behaviors such as restricting food intake, binge eating, using laxatives, or performing intense physical exercise in order to reduce body weight [8,9].

Disordered eating behaviors were prevalent in 15.9% of boys and 37.9% of girls with T1D in the United States [10], and similar results were obtained in Denmark, Canada, and Italy [8,11,12,13]. Among adults with T1D in the United States, 18% of men and 27% of women reported comorbid disordered eating behaviors [8]. These behaviors are not usually resolved without treatment [10] and are associated with poor glycemic control [14,15]. Therefore, disordered eating behaviors should be managed well among people with T1D.

Regular screening of disordered eating behaviors among people with T1D is recommended across international clinical guidelines [3,16]. There are three ways to measure disordered eating behaviors that are used globally: interview-based eating disorder examination for diagnosis, general questionnaires regarding disordered eating behaviors, and questionnaires specifically for people with diabetes. Interview-based eating disorder examination for diagnosis takes an average of one hour to be administered and needs specialists to interpret the results [17]. Therefore, it is not an optimal choice for routine screening, especially in developing countries with limited resources.

There are several validated general questionnaires regarding disordered eating behaviors that used in people with T1D such as the Eating Attitudes Test and the Eating Disorder Inventory-3 [18,19]. These questionnaires are used to screen for those in the general population who may have symptoms of eating disorders and need professional consultation. However, they may not be able to accurately estimate eating problems in a population with T1D. This is because diabetes management emphasizes carbohydrate counting and food intake. General measures of disordered eating may misidentify the appropriate level of attention on food intake for a person with T1D as a risky disordered eating behavior [20]. In addition, general questionnaires do not identify disordered eating behaviors that are unique to individuals with T1D, such as insulin restriction or omission [21].

There are three diabetes-specific questionnaires: the Diagnostic Survey for Eating Disorders, the Diabetes Eating Problems Survey (DEPS), and the Diabetes Eating Problem Survey-Revised (DEPS-R). The Diagnostic Survey for Eating Disorders was designed to diagnose eating disorders [22]. However, most disordered eating behaviors do not occur at a high enough frequency or severity to merit the formal diagnosis of eating disorders [23]. This suggests that using the Diagnostic Survey for Eating Disorders to screen for people with disordered eating behaviors may underestimate the prevalence of disordered eating behaviors among people with T1D [24].

The DEPS was the first diabetes-specific screening tool to measure disordered eating behaviors, but it was primarily validated among an adult sample, and 5 items in DEPS were redundant [25]. In 2010, DEPS was revised to the current 16-item Diabetes Eating Problem Survey-Revised (DEPS-R) [25]. The American Diabetes Association recommends using the DEPS-R to quickly screen for disordered eating behaviors [26]. It can be completed in 10 min and does not require specially trained clinicians to administer; thus, it has the potential to be applied in areas with limited healthcare resources [25]. Moreover, DEPS-R has been widely validated using a sample of both youths and adults, and type 1 and 2 diabetes in many countries, including Germany [27], Norway [28], Turkey [29], Spain [20], Italy [30], and Canada [31]. To date, it has been used as a screening tool for all ages of people with diabetes [30]. However, detailed translation procedures have not been reported although different language versions of DEPS-R were produced.

When preparing an instrument for use in a different culture, it is necessary to go beyond a direct language translation and also culturally adapt the instrument—in order to ensure that it will fit in the target cultural context (such as preferred eating habits, etc.). In a different culture or country, there may also be a different clinical context (such as optimal therapeutic regimen, etc.) [32]. Thus, proper context evaluation procedures need to be applied prior to language translation, otherwise, results may be attributed to context difference rather than differences in people or variables being measured [33]. However, rigorous and systematic efforts to evaluate the culture relevance or clinical scenario appropriateness of a scale before initiating translation procedures are rare.

In China, the incidence of T1D has increased 3.8-fold as compared to two decades ago, with China now ranking fourth in the world for the number of people with T1D [34]. There is a lack of Chinese data on the prevalence of disordered eating behaviors in this susceptible population. The specific measure of disordered eating behaviors in people with T1D is still lacking in China despite the fact that some general questionnaires have been used to screen for disordered eating behaviors in the Chinese general population [35]. DEPS-R is a diabetes-specific screening measure which has been recommended by the American Diabetes Association, recently updated, and validated in different cultures using both youths and adults. Therefore, the aims of this study were to translate the DEPS-R into Mandarin Chinese with cultural difference evaluation and also to test the psychometric properties of this new Chinese version using both youths and adults with T1D.

## 2. Methods

### 2.1. Study Design

This study included two consecutive phases: (1) context relevance evaluation and translation of the English version of DEPS-R into Mandarin Chinese, and (2) psychometric testing for youths and adults.

### 2.2. Phase 1: Context Relevance Evaluation and Instrument Translation

In August 2019, an expert panel was formed to evaluate context relevance of the DEPS-R given the current clinical and cultural context for Chinese people with T1D. The panel included an endocrinologist, two diabetes clinicians, two diabetes nurses, and a psychological therapist. All of them were bilingual, previously lived in an English-speaking country for at least one year and have been treating people with T1D in China for at least five years. Each aspect of the DEPS-R was briefly presented in English using a PowerPoint presentation by the first author, and all experts were then asked to comment on the cultural relevance of items one by one using the Context Relevance Index (CRI). The CRI was rated in the form of a four-point Likert scale where 1 represents “totally irrelevant”, 4 represents “totally relevant”. It is designed to evaluate the difference between the context in which an instrument was originally developed and the target context in terms of behavior, preference, religion, clinical regimen, and the healthcare system. All the items in the DEPS-R were determined to be relevant (≥3 points) to the Chinese context of people with T1D in China.

Based on Brislin’s (1986) adapted translation model [36], the English version was translated into Mandarin Chinese in the following three steps.

Step 1: Forward translation. Two Chinese-native translators who were fluent in both Mandarin Chinese and English completed the forward translation of the instrument independently. One was a graduate student in nursing who specialized in T1D and understood the objective of and concept behind the tool and the other was a cross-cultural bilingual expert who knew nothing about the instrument. The two translators provided additional explanation regarding two phrases, “eat to the point of spilling ketones in my urine” and “skip my next insulin dose”, which may not be clear to Chinese people with T1D. A meeting of the expert committee on translation (the two translators, two bilingual Chinese-native nurses with expertise in T1D, and a Chinese-native professor specializing in the English language) was then held. The goals were to compare the two forward-translated versions with the original version of the DEPS-R in terms of the denotation and connotation of the items and to discuss the most accurate and easily understood versions of terms. Two items were revised during the meeting. After that, consensus was achieved regarding the best wording in Mandarin Chinese. A copy was also provided to each expert for further review. This process produced a single forward translated DEPS-R (Chinese version) called the 1st C-DEPS-R.

Step 2: Back translation. The 1st C-DEPS-R was translated back into English by two other independent translators (B1-DEPS-R and B2-DEPS-R) who were fluent in both Mandarin Chinese and English and completely blinded to the original version of the instrument. Another meeting with the same expert committee followed the same procedure as step 1 to reach consensus on a back translated DEPS-R called B-DEPS-R.

Then, a panel of three bilingual nursing experts who knew DEPS-R well compared B-DEPS-R with both DEPS-R and 1st C-DEPS-R, identified problematic items, modified them after discussion, and eventually achieved consensus. For instance, Item 7 in the DEPS-R says, “I avoid checking my blood sugar when I feel like it is out of range” while in the B-DEPS-R it says, “I avoid testing my blood sugar when I feel like it is higher than normal blood sugar”. The back-translated item uses the inaccurate expression “higher than normal blood sugar” which could mislead subjects into not taking below-normal levels of blood sugar into account. Thus, the B- DEPS-R was completed and the 2nd C-DEPS-R was derived.

Step 3: Evaluation of translation equivalence. The items of the 2nd C-DEPS-R were scored one by one using the Translation Validity Index (TVI), which was adapted from the Content Validity Index [33] in the form of a four-point Likert scale where 1 represents “totally different” and 4 represents “equivalent”. In this way, systematic judgments were obtained from three new bilingual experts concerning the translation equivalence between the original English version of the DEPS-R and the 2nd C-DEPS-R. These experts were selected based on their rich experience in clinical diabetes care, translating well-established English instruments into Mandarin Chinese, diabetes instrument development, and English language translation. After two rounds of these processes (expert judgment, modification and reconciliation, and then expert judgment again), the instrument translation procedure was completed successfully. The translation procedure is shown in Figure 1.

### 2.3. Testing of the Psychometric Properties

A cross-sectional survey design was used in this study to evaluate psychometric properties including internal consistency reliability, test-retest reliability, distribution of scores, and convergent validity among youths and adults with T1D. It was conducted from July 2019 to August 2019. The Strengthening the Reporting of Observational Studies in Epidemiology (STROBE) checklist was used for this study.

### 2.4. Setting and Sample

The research site was located at the Diabetes Center in the capital city of Hunan province, China, the place where the largest number of people with T1D in the local area was registered (deleted for double-blind review). Inclusion criteria for the tests of the psychometric properties among youths and adults were almost the same and included: (1) diagnosed with T1D for at least three months; (2) receiving treatment with insulin; (3) able to read and speak in Mandarin Chinese; and (4) aged 8~18 years old (for testing among youths) or aged 18 years old or more (for testing among adults). Participants were excluded if they had (1) serious diabetes complications including hypoglycemia, neuropathy, kidney disease, heart disease, eye disease, or amputation, among others; (2) untreated Attention Deficit Hyperactivity Disorder; or (3) other serious physical or psychiatric conditions such as thyroid disease, asthma, and hypertension.

### 2.5. Recruitment and Data Collection

This study was from the secondary data analysis of a camp-based education program with a quasi-experimental design [37]. Two trained diabetes educators working at the research site either contacted eligible people, including the parents of those under 18 years of age, by telephone after reviewing their medical records or directly invited them during a regular clinic visit. These eligible people were invited to participate after the purpose, benefits, and risks of the study were fully disclosed by two diabetes educators. If people were interested in learning more about the study, a trained research assistant further described the study in detail and obtained written informed consent from the participants, and consent from parents of those under 18 years of age.

They were then invited to complete the self-reported measures and anthropometric measures in a quiet room at the research site, which included examinations of diabetes-specific disordered eating behaviors, anxiety, body mass index, and glycated hemoglobin. Fifteen youths and 12 adults retook the examinations of disordered eating behaviors using C-DEPS-R at 3 weeks (±3 days). In terms of the self-reported measures, the research assistants were available to answer questions, and they checked each questionnaire to prevent unintentional missing items or incomplete pages. Young participants were given the option to complete the forms by themselves or to have the research assistants read the items and fill in their answers for them. As for anthropometric measures, the height and weight of all the participants in light clothing and shoeless was measured by trained nurses using portable stadiometers and calibrated balance scales, respectively. Height was measured to the nearest 0.5 cm and weight was taken with a precision of 0.1 kg.

### 2.6. Measures

Demographic and T1D-related information including age, sex, weight, height, duration of T1D, and treatment modality was collected by an investigator-designed Demographic and Clinical Data Sheet.

Diabetes-specific disordered eating behaviors were evaluated by the Chinese version of the Diabetes Eating Problem Survey-Revised (C-DEPS-R), which is a 16-item diabetes-specific self-reported questionnaire. Items are scored on a six-point Likert scale where “0” represents “never,” and “5” represents “always.” Total C-DEPS-R score results from the sum of the scores of 16 items. The total score could range from 0 to 80, with a higher score indicating more disordered eating behaviors. A cutoff point of 20 indicates a high risk for disordered eating behaviors with an overall range of 0–80. The scoring among youths and adults was the same. The original DEPS-R has been shown to have a good internal consistency (Cronbach’s alpha = 0.86) and construct validity (significant positive correlations with body mass index (BMI) [*r* = 0.412, *p* < 0.01], HbA1c levels [*r* = 0.303, *p* < 0.01], and age [*r* = 0.194, *p* < 0.01] in a sample of the pediatric population with T1D) [25].

Anxiety (including state anxiety and trait anxiety) was evaluated by the youth and adult versions of the Chinese version of the State-Trait Anxiety Inventory (C-STAI). The adult version is composed of two subscales: The State Anxiety Subscale (STAI-S) and the Trait Anxiety Subscale (STAI-T). STAI-S includes 20 items; respondents are asked to assess the intensity of their current feelings on a 4-point scale, ranging from 1 = not at all to 4 = very much so. STAI-T also includes 20 items that are rated on a different 4-point scale, ranging from 1 = almost never to 4 = almost always, and participants are asked to choose the statement that most closely describes the frequency of their feelings. Higher scores indicate greater anxiety in both subscales, and a cutoff point of 40 has been suggested to indicate clinically significant state or trait anxiety. The Cronbach’s alpha of the adult version of the C-STAI ranges from 0.86 to 0.95 [38]. The youth version of C-STAI, intended for people aged 8–17 years, is the most widely used instrument for measuring anxiety in youths. It is similar to the adult version except that the measure is easier to understand and can be administered verbally to youths. In previous studies, Cronbach’s alpha of the youth version of the C-STAI for state and trait anxiety were reported as 0.84 and 0.87, respectively [39].

BMI was calculated by dividing body mass in kilograms by height in meters squared (kg/m^2^). For adults, a BMI of 27.9 kg/m^2^ and below is considered underweight/normal, and a BMI of 28 kg/m^2^ and above is considered overweight or obese [40]. For youths, a BMI below the 85th percentile is considered underweight and normal weight and a BMI at or above the 85th percentile is defined as overweight and obese for youths of the same age and sex [41]. Weight status was, therefore, categorized into two groups: underweight/normal weight or overweight/obesity.

Glycated Hemoglobin (HbA1c) is an indicator of metabolic control over the past three months. The HbA1c results for all participants came from their medical records over the past three months. Higher values reflect poorer metabolic control. The cutoff point value of HbA1c is 7% for adults (aged ≥18) and 7.5% for youths (aged under 18) which are the care goals for people with T1D [42].

### 2.7. Reliability and Validity Evaluation

Reliability refers to the extent of the homogeneity of all the items including internal consistency reliability and test-retest reliability. Internal consistency reliability is generally measured by Cronbach’s alpha [43]. A commonly-accepted rule is that an alpha of 0.7 indicates acceptable reliability and 0.8 or higher indicates good reliability. Test-retest reliability is generally measured by the Pearson correlation coefficient, which is acceptable when over 0.85 at three weeks.

Validity evaluates the accuracy of an instrument [43]. The validity of this Chinese version of the scale was tested using three aspects: exploratory factor analysis, distribution of scores, and convergent validity.

Exploratory factor analysis uncovers the underlying structure of 16-item C-DEPS-R. Distribution of scores refers to the frequency at which each option is selected by participants and can be used to measure the sensitivity of the C-DEPS-R to variation. Convergent validity refers to how closely the C-DEPS-R is related to other measures that are theoretically correlated with C-DEPS-R and disordered eating behaviors including STAI, BMI, and HbA1c. The values of convergent validity were obtained through the strength of the relationship between STAI (or BMI, HbA1c) and the scores from the C-DEPS-R [44].

### 2.8. Data Analyses

The Statistical Package for Social Science (SPSS Inc., Chicago, IL, USA) Version 23.0 for Windows was used for all data analysis. The conventional level of significance was 0.05. Data were analyzed using the same analysis method for both the youth and adult sample. Descriptive statistics were calculated for all study variables.

The internal consistency reliability and test-retest reliability of the C-DEPS-R was analyzed using Cronbach’s alpha and Pearson correlation index respectively. For construct validity, since the factorial structure of the DEPS-R has not been reported before, it is recommended to perform exploratory factor analysis when the DEPS-R is revised or adapted to the other culture [25]. The principal component and Varimax rotation were used to conduct exploratory factor analysis, and the sample adequacy was assessed by the Kaiser–Meyer–Olkin measure. Factors with eigenvalues higher than 1.0 and items with loadings greater than 0.4 were accepted. Confirmatory factor analysis has not been conducted due to the limited sample size. The distribution of scores was obtained by counting the frequency at which each option in the C-DEPS-R was selected. To indicate convergent validity, correlations between the C-DEPS-R with STAI (or BMI, HbA1c) were analyzed using Pearson and Spearman correlation.

## 3. Results

### 3.1. Context Relevance and Translation Equivalence Evaluation

In terms of context relevance evaluation, all items on this scale were scored “3 = relevant or 4 = totally relevant” by all experts. Regarding translation equivalence, a satisfactory consensus was obtained with 88% of items in C-DEPS-R rated with score 4 and 100% items rated as 3 or 4 by all expert on TVIs.

### 3.2. Demographics and T1D-Related Characteristics of Participants

Of the 100 youths and 70 adults with T1D who participated, 11 youths (11%) and 9 adults (12.8%) were excluded because they did not complete the questionnaire. Consequently, a total of 89 youths (89%) and 61 adults (87.2%) were ultimately included in the study. For both the youth and adult groups, there was no significant difference in demographics (age, sex) and clinical characteristics (diabetes duration) between the participants included in this analysis and those who were not included.

Regarding the 89 youths with T1D, the mean age was 13.0 (SD = 2.5) years; 59.3% (*n* = 38) were school-aged children (8–12 years) and 40.7% (*n* = 51) were adolescents (13–17 years). The majority of youths were female (62.9%, *n* = 56). The mean T1D duration was 4.8 (SD = 3.1) years, and 59.6% (*n* = 53) had had diabetes for less than five years. In addition, 74.2% of youths (*n* = 66) did not use insulin pump therapy. The mean HbA1c was 8.5% (SD = 2.3%), of which 39.3% (*n* = 35) had abnormal HbA1c (>7.5%). The mean BMI was 18.8 (SD = 2.7). The mean score of STAI-S was 40.5 (SD = 11.7) and the mean score of STAI-T was 41.2 (SD = 10.1). The mean score of the C-DEPS-R was 21.0 (SD = 9.7) with a range from 4–44. More than one-third (*n* = 35, 39.3%) had C-DEPS-R scores over 20.

Regarding the 61 adults with T1D, the mean age was 33.0 (SD = 14.5) and nearly half of the adults (47.5%, *n* = 29) were male. The mean T1D duration was 8.3 (SD = 5.0) years and 24.6% (*n* = 15) had had diabetes for less than five years. In addition, 85.2% of the adults (*n* = 52) did not use insulin pump therapy. The mean HbA1c was 7.6% (SD = 1.5%) with a range of 5.4–12.3%. Abnormal HbA1c of less than 7.0% was noted in 59% (*n* = 36) of the adults. Their mean BMI was 21.4 (SD = 2.5) kg/m^2^. The mean score of the STAI-S was 42.6 (SD = 9.9) and the mean score of the STAI-T was 42.5 (SD = 11.1). The mean score of the C-DEPS-R was 20.7 (SD = 11.1) and 45.9% (*n* = 28) had C-DEPS-R scores over 20. The descriptions of both youth and adult participants with T1D are included in Table 1.

### 3.3. Test-Retest Reliability

Among the 15 youths and 12 adults who completed the questionnaire packet on two occasions, the Pearson correlation coefficient for the C-DEPS-R was 0.916 and 0.873, respectively.

### 3.4. Internal Consistency Reliability

Cronbach’s alpha for the C-DEPS-R among youths and adults with T1D was 0.85 and 0.78, respectively.

### 3.5. Construct Validity

#### 3.5.1. Exploratory Factor Analysis

The Kaiser–Meyer–Oklin value was 0.702 and 0.790 among youths and adults with T1D, respectively, which exceeded the minimum recommended value of 0.60, and Bartlett’s test of sphericity reached statistical significance, *p* < 0.001. The exploratory factor analysis showed that a rotated factor solution for the C-DEPS-R contained three factors with eigenvalues greater than 1.0, and accounting for 53.5% and 56.8 of variance among youths and adults with T1D, respectively. Factor 1 contained nine items (2, 3, 4, 5, 7, 12, 13, 14 and 15), Factor 2 contained four items (1, 6, 11 and 16) and Factor 3 contained three items (8, 9 and 10). All item loadings were greater than 0.4 among youths and adults with T1D.

#### 3.5.2. Inter-Correlations and Distributions of Scores

Regarding youths with T1D, the correlation coefficients between the scores of each item and the total score of the C-DEPS-R were statistically significant (*p* < 0.001), ranging from 0.329 to 0.655. Regarding the adults with T1D, the correlation coefficients between the scores of each item and the total score of the C-DEPS-R were statistically significant (*p* < 0.001), ranging from 0.424 to 0.705. These correlations are shown in Table 2.

The mean score, standard deviation, frequency of each option, and the composition ratio of the 16 items among youths and adults are displayed in Table 3 and Table 4, respectively. Regarding youths with T1D, the rate of the highest score for each item was 0.0–19.1% and the lowest score was 1.1–90.0%. However, for 13 of the 16 items, the rate of the lowest score for all the items in the C-DEPS-R exceeded 15%. In addition, item 12, “Other people tell me to take better care of my diabetes”, had the highest rate of scores over 15%. Regarding adults with T1D, the rate of the lowest score for each item was 3.3–87% and the highest score was 0.0–13.1%. For the same 13 items, the rate of the lowest score for all the items in the C-DEPS-R exceeded 15% (the three items which the lowest score did not exceed 15% were 3%, 12%, and 14%).

#### 3.5.3. Convergent Validity

For convergent validity regarding youths with T1D, state and trait anxiety were significantly and positively associated with disordered eating behaviors (*r* = 0.310, *p* = 0.003; *r* = 0.313, *p* = 0.003). BMI and HbA1c were also significantly and positively associated with disordered eating behaviors (*r* = 0.255, *p* = 0.016; *r* = 0.459, *p* = 0.000).

In terms of adults with T1D, state and trait anxiety were significantly and positively associated with disordered eating behaviors (*r* = 0.373, *p* = 0.003; *r* = 0.313, *p* = 0.013) while BMI and HbA1c were not significantly associated with disordered eating behaviors (*r* = 0.083, *p* = 0.523; *r* = 0.215, *p* = 0.097). These associations are shown in Table 5.

## 4. Discussion

This study is the first to report on the context relevance evaluation and detailed translation process of the DEPS-R into a different context or language. In this study, a consensus of context relevance was achieved with high CRI to quantify the lack of significant context difference, laying the foundation for the subsequent translation. The translation process followed the Brislin’s translation model and TVI was used to quantify the quality of the translation, assuring fidelity to the original instrument.

The psychometric testing results provided evidence of adequate reliability of the C-DEPS-R among youths and adults with T1D in China. In terms of youths, the Cronbach coefficient of 0.85 in the C-DEPS-R showed excellent internal consistency [43], which is in line with the findings from Norwegian and Turkish youths with T1D [28,29]. Internal consistency was also found to be good in terms of the adults with a Cronbach coefficient of 0.78, which was slightly lower than Norwegian Cronbach coefficient of 0.84 among adults aged 18-79 years [44]. Test–retest reliability for the C-DEPS-R among youths and adults indicated good temporal stability.

The results of the exploratory factor analysis showed a three-factor solution. This structure was in line with the results of the DEPS-R translated version in Italy [30]. This study is the first to report the ceiling and floor effects of the items of the C-DEPS-R. The proportion of the lowest score in most of the items in the C-DEPS-R is more than 15% among youths and adults with T1D, indicating floor effects for these items [45]. It is likely that extreme items are missing on the lower end of the scale, indicating limited content validity [46]. In addition, regarding youths with T1D, item 12 of the C-DEPS-R showed ceiling effects, that is, youths tended to get the highest scores in item 12. This might be because they were recruited from the Diabetes Center of our university, which provides diabetic education about management of T1D to both youths and their parents. Thus, the parents of youths with T1D in our study might provide more suggestions and requirements for diabetes management, including diet management.

The convergent validity of the C-DEPS-R among youths and adults with T1D was demonstrated. HbA1c levels (indicating glycemic control) and BMI were positively correlated with the total scores of the C-DEPS-R in youths, which is consistent with previous findings in youths with T1D [47,48,49]. In addition, the association between the disordered eating behaviors and anxiety observed in this study is in line with the results of previous studies [50,51]. The relationships between BMI, anxiety, and disordered eating behaviors echo the evidence that more concerns about body weight and shape could lead to more anxiety, which is associated with the onset of disordered eating behaviors [10,49]. In the adults there was an association between disordered eating behaviors and anxiety, which is consistent with other studies [51]. However, HbA1c levels and BMI were not associated with disordered eating behaviors, which was inconsistent with other studies of adults with T1D [13,30,44]. This might be because the sample size of adults was not large enough to identify the significant relationship between these variables.

Previous studies among youths and adults with T1D in different countries defined a score of DEPS-R ≥20 as an indicator of high risk for disordered eating behaviors [25,27]. The proportion of high risk of disordered eating behaviors in youths in this study (39.3%) is higher than adolescents with TID aged 13–17 in the United States (15.0%) [52] and Turkish children and adolescents aged 9–18 (25.0%) [29]. Nearly half of the adults (45.9%) had high risk of disordered eating behaviors, a higher rate than adults aged 18–28 in in the United States (22.0%). This might be because little attention has been paid to eating disorders in people with T1D in China, as there were no previous studies in China on this topic to our knowledge. In addition, more research is needed to explore the optimal cue-off score of the C-DEPS-R in different contexts and populations.

This study has several limitations. First, the psychometric testing of the C-DEPS-R did not include tests of equivalence based on responses of bilingual people in both language versions because it was difficult to recruit enough Mandarin-speaking people with T1D who could also answer the scale in English. Second, confirmatory factor analysis was not conducted due to the limited sample size from the parent study, but the other psychometric properties were reported on in this study for both youths and adults.

Despite its limitations, this study has several important clinical and research implications. A short, self-administered, reliable, and valid tool, the 16-item diabetes-specific C-DEPS-R, could be used to screen for disordered eating behaviors in youths and adults with T1D in China according to international guidelines. The C-DEPS-R could help clinicians determine whether a more extensive assessment is needed among youths and adults with T1D, especially when BMI, or HbA1c is also high. Further studies exploring the optimal cut-off score of the C-DEPS-R in different contexts and different populations might be necessary. Furthermore, the comparison of disordered eating behaviors in youths and adults with T1D between the Chinese population and other populations in different countries could be possible through the use of different language versions of the DEPS-R, which could provide insight into the cultural variability of disordered eating behaviors. In addition, studies with larger sample sizes and multi-site populations are needed to provide more evidence on the validity of the C-DEPS-R in China. Finally, the context relevance index was conceptualized before initiating any instrument translation. We recommend its use to evaluate applicability of an instrument to a target context which is different than the context in which the instrument was originally developed.

## 5. Conclusions

This study was the first to evaluate the context relevance and translation fidelity of the DEPS-R by including quantified indexes to guarantee the rigor and quality of the instrument translation. The C-DEPS-R is a reliable and valid tool to screen for disordered eating behaviors in youths and adults with T1D; its reliability was also demonstrated in Chinese adults. This study reported a high proportion of disordered eating behaviors among both youths and adults with T1D, indicating a need for special attention from healthcare professionals and researchers in China.

## Figures and Tables

**Figure 1 ijerph-18-03450-f001:**
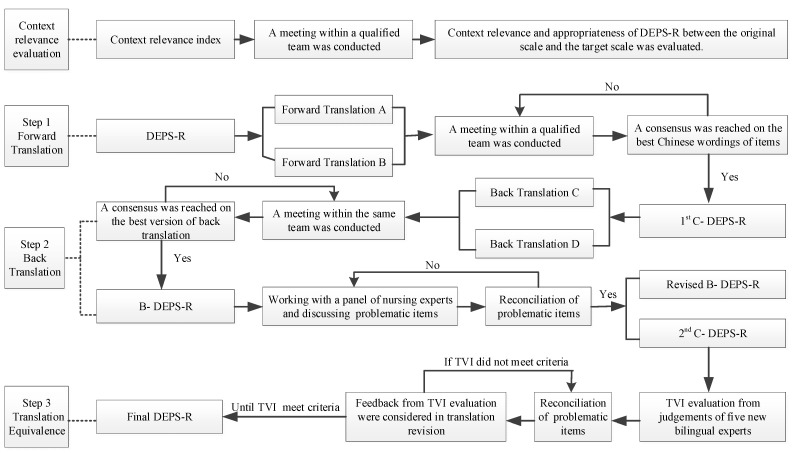
Context relevance evaluation and instrument translation procedure of DEPS-R. Note. DEPS-R, the original English version of the Diabetes Eating Problem Survey-Revised; C-DEPS-R, the translated Chinese version of the Diabetes Eating Problem Survey-Revised; B-DEPS-R, the back-translated version of the Diabetes Eating Problem Survey-Revised (in English); TVI, translation validity index.

**Table 1 ijerph-18-03450-t001:** Demographics and type 1 diabetes (T1D)-related characteristics of youths (*n* = 89) and adults (*n* = 61).

	Youths (*n* = 89)		Adults (*n* = 61)	
	Mean ± SD	N (%)	Mean ± SD	N (%)
Age (years)	12.97 ± 2.48		32.98 ± 14.47	
children (8–12)		38 (42.7%)		
adolescents (13–17)		51 (57.3%)		
Sex				
male		33 (37.1%)		29 (47.5%)
female		56 (62.9%)		32 (52.5%)
Diabetes duration (years)	4.83 ± 3.11		8.30 ± 4.97	
<5		53 (59.6%)		15 (24.6%)
≥5		36 (40.4%)		46 (75.4%)
Insulin pump therapy				
yes		23 (25.8%)		9 (14.8%)
no		66 (74.2%)		52 (85.2%)
HbA1c (%)	8.54 ± 2.32		7.55 ± 1.48	
normal		35 (39.3%)		25 (41.0%)
abnormal		54 (60.7%)		36 (59.0%)
BMI (kg/m^2^)	18.83 ± 2.69		21.44 ± 2.16	
underweight or normal		83 (93.3%)		53 (86.9%)
overweight or obese		6 (6.7%)		8 (13.1%)
STAI				
STAI-S	40.51 ± 11.71		42.59 ± 9.85	
STAI-T	40.15 ± 10.12		42.49 ± 11.07	
C-DEPS-R	20.96 ± 9.73		20.65 ± 11.07	
≤20		54 (60.7%)		33 (54.1%)
>20		35 (39.3%)		28 (45.9%)

Note: SD, standard deviation; HbA1C, glycated hemoglobin; BMI, body mass index; STAI, State-Trait Anxiety Inventory; STAI-S, the State Anxiety Subscale; STAI-T, the Trait Anxiety Subscale; C-DEPS-R, the Chinese version of the Diabetes Eating Problem Survey-Revised.

**Table 2 ijerph-18-03450-t002:** The correlations between items and total scores of C-DEPS-R of youths (*n* = 89) and adults (*n* = 61).

Items		Youths		Adults	
		Items Scores Mean (SD)	*r*	Items Scores Mean (SD)	*r*
1	Losing weight is an important goal to me.	1.80 (1.69)	0.411 **	1.67 (1.61)	0.632 **
2	I skip meals and/or snacks.	0.83 (0.95)	0.412 **	0.80 (0.98)	0.465 **
3	Other people have told me that my eating is out of control.	1.95 (1.26)	0.591 **	1.95 (1.19)	0.665 **
4	When I overeat, I don’t take enough insulin to cover the food.	1.44 (1.27)	0.406 **	1.54 (1.23)	0.673 **
5	I eat more when I am alone than when I am with others.	1.28 (1.28)	0.389 **	1.15 (1.16)	0.452 **
6	I feel that it’s difficult to lose weight and control my diabetes at the same time.	1.39 (1.38)	0.655 **	1.57 (1.44)	0.625 **
7	I avoid checking my blood sugar when I feel like it is out of range.	0.84 (1.22)	0.613 **	0.74 (1.12)	0.633 **
8	I make myself vomit.	0.22 (0.64)	0.329 **	0.21 (0.69)	0.572 **
9	I try to keep my blood sugar high so that I will lose weight.	0.22 (0.66)	0.356 **	0.33 (0.85)	0.465 **
10	I try to eat to the point of spilling ketones in my urine.	1.22 (1.47)	0.400 **	1.16 (1.50)	0.413 **
11	I feel fat when I take all of my insulin.	1.58 (1.65)	0.605 **	1.48 (1.61)	0.622 **
12	Other people tell me to take better care of my diabetes.	3.19 (1.30)	0.402 **	3.07 (1.34)	0.507 **
13	After I overeat, I skip my next insulin dose.	0.27 (0.69)	0.480 **	0.25 (0.68)	0.587 **
14	I feel that my eating is out of control.	1.96 (1.34)	0.637 **	2.07 (1.20)	0.620 **
15	I alternate between eating very little and eating huge amounts.	1.90 (1.38)	0.608 **	1.92 (1.46)	0.705 **
16	I would rather be thin than to have good control of my diabetes.	0.74 (1.23)	0.505 **	0.75 (1.35)	0.424 **
Total score Mean (SD)		20.96 ± 9.73		20.65 ± 11.07	

** *p* < 0.001; Note: SD, standard deviation.

**Table 3 ijerph-18-03450-t003:** Distribution of item scores on the C-DEPS-R among youths (*n* = 89).

Items	Score Mean (SD)	Option N (%)					
		Never	Rarely	Sometimes	Often	Usually	Always
1	1.80 (1.69)	26 (29.2%)	20 (22.5%)	12 (13.5%)	11 (12.3%)	9 (10.1%)	11 (12.4%)
2	0.83 (0.95)	38 (42.7%)	34 (38.2%)	12 (13.5%)	3 (3.4%)	2 (2.2%)	0 (0.0%)
3	1.95 (1.26)	11 (12.4%)	28 (31.4%)	20 (22.5%)	18 (20.2%)	9 (10.1%)	3 (3.4%)
4	1.44 (1.27)	25 (28.0%)	32 (35.9%)	15 (16.9%)	11 (12.4%)	3 (3.4%)	3 (3.4%)
5	1.28 (1.28)	27 (30.3%)	31 (34.9%)	14 (15.7%)	7 (7.9%)	8 (9.0%)	2 (2.2%)
6	1.39 (1.38)	35 (39.3%)	22 (24.7%)	14 (15.7%)	11 (12.4%)	6 (6.7%)	1 (1.1%)
7	0.84 (1.22)	46 (51.7%)	26 (29.2%)	2 (2.2%)	10 (11.3%)	3 (3.4%)	2 (2.2%)
8	0.22 (0.64)	76 (85.4%)	8 (9.0%)	3 (3.4%)	2 (2.2%)	0 (0.0%)	0 (0.0%)
9	0.22 (0.66)	80 (90.0%)	6 (6.7%)	2 (2.2%)	1 (1.1%)	0 (0.0%)	0 (0.0%)
10	1.22 (1.47)	41 (46.0%)	14 (15.7%)	15 (16.9%)	12 (13.5%)	3 (3.4%)	4 (4.5%)
11	1.58 (1.65)	34 (38.2%)	14 (15.7%)	12 (13.5%)	13 (14.6%)	9 (10.1%)	7 (7.9%)
12	3.19 (1.30)	1 (1.1%)	7 (7.9%)	18 (20.2%)	20 (22.5%)	26 (29.2%)	17 (19.1%)
13	0.27 (0.69)	72 (81.0%)	13(14.6%)	1 (1.1%)	2 (2.2%)	1 (1.1%)	0 (0.0%)
14	1.96 (1.34)	16 (14.7%)	37 (24.7%)	44 (29.4%)	24 (16%)	18 (12%)	5 (3.3%)
15	1.90 (1.38)	14 (15.7%)	25 (28.1%)	19 (21.3%)	20 (22.5%)	9 (10.1%)	2 (2.2%)
16	0.74 (1.23)	54 (60.7%)	17 (19.1%)	12 (13.5%)	3(3.4%)	0 (0.0%)	3 (3.4%)

Note: SD, standard deviation.

**Table 4 ijerph-18-03450-t004:** Distribution of item scores on the C-DEPS-R among adults (*n* = 61).

Items	Score Mean (SD)	Option N (%)					
		Never	Rarely	Sometimes	Often	Usually	Always
1	1.67 (1.61)	22 (36.1%)	9 (14.7%)	10 (16.4%)	10 (16.4%)	7 (11.5%)	3 (4.9%)
2	0.80 (0.98)	31 (50.8%)	16 (26.2%)	9 (14.8%)	5 (8.2%)	0 (0.0%)	0 (0.0%)
3	1.95 (1.19)	6 (9.8%)	20 (32.8%)	12 (19.7%)	17 (27.9%)	6 (9.8%)	0 (0.0%)
4	1.54 (1.23)	14 (23.0%)	17 (27.9%)	19 (31.1%)	6 (9.8%)	4 (6.6%)	1 (1.6%)
5	1.15 (1.16)	23 (37.7%)	17 (27.9%)	13 (21.3%)	5 (8.2%)	3 (4.9%)	0 (0.0%)
6	1.57 (1.44)	18 (29.5%)	16 (26.2%)	10 (16.4%)	10 (16.4%)	5 (8.2%)	2 (3.3%)
7	0.74 (1.12)	35 (57.4%)	16 (26.2%)	4 (6.6%)	4 (6.6%)	1 (1.6%)	1 (1.6%)
8	0.21 (0.69)	53 (87.0%)	6 (9.8%)	0 (0.0%)	1 (1.6%)	1 (1.6%)	0 (0.0%)
9	0.33 (0.85)	50 (82.0%)	6 (9.8%)	3 (4.9%)	0 (0.0%)	2 (3.3%)	0 (0.0%)
10	1.16 (1.50)	30 (49.1%)	14 (23.0%)	4 (6.6%)	3 (4.9%)	10 (16.4%)	0 (0.0%)
11	1.48 (1.61)	26 (42.6%)	9 (14.8%)	9 (14.8%)	8 (13.1%)	6 (9.8%)	3 (4.9%)
12	3.07 (1.34)	2 (3.3%)	7 (11.5%)	11 (18.0%)	14 (23.0%)	19 (31.1%)	8 (13.1%)
13	0.25 (0.68)	51 (83.6%)	7 (11.5%)	2 (3.3%)	0 (0.0%)	1 (1.6%)	0 (0.0%)
14	2.07 (1.20)	6 (9.8%)	14 (23.0%)	20 (32.8%)	13 (21.3%)	7 (11.5%)	1 (1.6%)
15	1.92 (1.46)	12 (19.6%)	14 (23.0%)	15 (24.6%)	11 (18.0%)	5 (8.2%)	4 (6.6%)
16	0.75 (1.35)	42 (68.8%)	6 (9.8%)	5 (8.2%)	4 (6.6%)	2 (3.3%)	2 (3.3%)

Note: SD, standard deviation.

**Table 5 ijerph-18-03450-t005:** Measures and correlations with C-DEPS-R of youths (*n* = 89) and adults (*n* = 61).

Measures	Scores (Mean ± SD)	C-DEPS-R
STAI-S		
Youths	40.51 ± 11.71	*r* = 0.310; *p* = 0.003
Adults	42.59 ± 9.85	*r* = 0.373; *p* = 0.003
STAI-T		
Youths	41.15 ± 10.12	*r* = 0.313; *p* = 0.003
Adults	42.49 ± 11.07	*r* = 0.317; *p* = 0.013
BMI		
Youths	18.83 ± 2.69	*r* = 0.255; *p* = 0.016
Adults	21.44 ± 2.16	*r* = 0.083; *p* = 0.523
HbA1c		
Youths	8.54 ± 2.32	*r* = 0.459; *p* = 0.000
Adults	7.55 ± 1.48	*r* = 0.215; *p* = 0.097

Note: SD, standard deviation; STAI-S, State Anxiety Subscale; STAI-T, Trait Anxiety Subscale; BMI, body mass index; HbA1C, glycated hemoglobin.

## Data Availability

The data presented in this study are available on request from the corresponding author. The data are not publicly available due to privacy.

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
