# Peer review of "Instrument Context Relevance Evaluation, Translation, and Psychometric Testing of the Diabetes Eating Problem Survey-Revised (DEPS-R) among People with Type 1 Diabetes in China"

_ijerph, 2021, doi:10.3390/ijerph18073450_

Round 1
Reviewer 1 Report
Aim of this paper is to evaluate the context relevance and translation fidelity of the Diabetes Eating Problem Survey-Revised (DEPS-R) into Mandarin Chinese and then test its psychometric properties among youths and adults with type 1 diabetes in China.
The study was well conducted and the data presented are very clear and effective.
Before publication there are a few points that need to be clarified:
- it seems to me that the abstract exceeds 200 words which is the limit of the journal. The authors should shorten the abstract and better summarise the presentation and discussion of the data.
- why was the DEPS-R chosen over other tests in the literature?
- the diet of a Chinese person is different from that of a Western youth or adult... was this aspect taken into account by the authors? could it not have influenced the validation of the test?
Reviewer 2 Report
I would like to congratulate the Authors for the quality of the manuscript they submitted to the Journal. This paper not only provides important information regarding factors contributing to the successful treatment of T1DM, but also provides a detailed description on how to appropriately adapt a research tool in medical sciences.
Reviewer 3 Report
Dear authors:
This manuscript is writing about type-1 diabetes with an eating program survey. Why do you want to know this?
“Instrument context relevance evaluation, translation, and psychometric testing of the Diabetes Eating Problem Survey-Revised (DEPS-R) among people with type 1 diabetes in China.”
1.Abstract need short (<399 words).
2.Figure 1: Need distinguished yes or no. Step 1, Step 2, and Step 3.
3.Need to show school department.
Table 1. Demographics and T1D-related characteristics of youths and adults.
Table 2. The correlations between items and total scores of C-DEPS-R of youths and adults.
How to measure total scores of C-DEPS-R of youths and adults? Mean±SD.?
Table 3. Distribution of item scores on the C-DEPS-R among youths (n=89).
Need Mean ± SD. or Mean ± SEM.?
Table 4. Distribution of item scores on the C-DEPS-R among adults (n=61).
Need Mean ± SD. or Mean ± SEM.?
Table 5. Measures and correlations with C-DEPS-R of youths and adults.
N=?
Table 6Statistical comparison of demographics and T1D-related characteristics of youths and adults.
N=?
Reviewer 4 Report
Abstract: nothing to add.
Introduction.
Authors highlight the importance of the DEPS-R and the necessity of both translation to mandarin and adaptation to the cultural context in China.
Methods.
Authors conducted the study in two phases: 1) Context relevance and Instrument translation which were divided in 3 steps and 2) Psychometric testing, including internal consistency reliability, test-retest reliability, exploratory factor analysis, distributions of scores, convergent validity, and discriminant validity.
I wonder why authors choose an EFA instead of CFA because it is an established test widely used in other cultural contexts. Perhaps they should be used both approaches: 1) to confirm previous structure and then, if previous structure is not confirmed, 2) exploring.
However, this is not mandatory, it is only an opinion.I did not find the method followed by authors: Maximum likelihood, Principal Component analysis… Please, clarify this issue.
Results and discussion are in correspondence with objectives and methods.
Round 2
Reviewer 3 Report
Dear Authors:
Need to write your city and country.
Abstract deleted (1), (2), (3), (4).
Table 2. Citied **p-value < 0.0001. Or **p< 0.001.
Table 5. what is +? in SD, AND MEASURE GROUPS. And Table 6.
That is it.
Good day
